# Increased Expression Levels of Thermophilic Serine Protease TTHA0724 through Signal Peptide Screening in *Bacillus subtilis* and Applications of the Enzyme

**DOI:** 10.3390/ijms242115950

**Published:** 2023-11-03

**Authors:** Yiwen Xu, Xiaoran Xuan, Renjun Gao, Guiqiu Xie

**Affiliations:** 1School of Pharmaceutical Sciences, Jilin University, Changchun 130021, China; xuyw21@mails.jlu.edu.cn; 2Key Laboratory for Molecular Enzymology and Engineering of Ministry of Education, School of Life Science, Jilin University, Changchun 130021, China; xuanxr19@mails.jlu.edu.cn (X.X.); gaorj@jlu.edu.cn (R.G.)

**Keywords:** *B. subtilis*, thermophilic serine protease, heterologous expression, detergent

## Abstract

The thermostable protease TTHA0724 derived from *Thermus thermophilus* HB8 is an ideal industrial washing enzyme due to its thermophilic characteristics; although it can be expressed in *Escherichia coli* via pET-22b, high yields are difficult to achieve, leading to frequent autolysis of the host. This paper details the development of a signal peptide library in the expression system of *B. subtilis* and the optimization of signal peptides for enhanced extracellular expression of TTHA0724. When *B. subtilis* was used as the host and the optimized signal peptide was used, the expression level of TTHA0724 was 16.7 times higher compared with *E. coli*. *B. subtilis* as an expression host does not change the characteristics of TTHA0724. The potential application fields of TTHA0724 are studied. TTHA0724 can be used as a detergent additive at 60 °C, which can sterilize and eliminate mites while thoroughly cleaning protein stains. Soybean meal enzymatic hydrolysis with TTHA0724 at a high temperature produced a higher content of antioxidant peptides. These results indicate that TTHA0724 has great potential for industrial applications.

## 1. Introduction

Microbially derived proteases account for two-thirds of the world’s commercial proteases. Proteases not only play important roles in metabolic activities but also have great application value in industry. Proteolysis can be carried out by chemical or enzymatic processes. Chemical processes, including alkaline or acidic hydrolysis, often produce a large amount of wastewater. Enzymatic processes are mild and can avoid side reactions. The genus Bacillus is essential for the commercially important alkaline protease, which is active in the alkaline pH range between 9 and 11 [1]. Serine proteases account for a third of the enzyme market and are used simultaneously in pharmaceutical, food, and various other industries, including the production of cheese [2,3], detergents [4], and leather depilation [5]. TTHA0724 from *T. thermophilus* HB8 belongs to the S8 serine protease family, and it is a thermophilic enzyme. Therefore, TTHA0724 can be better tolerated than mesophilic enzymes and can withstand more harsh industrial production environments. TTHA0724 comprises 434 amino acids, the first 28 of which form a possible propeptide [6,7]. TTHA0724 contains propeptide, and it can only become an active mature protease after the propeptide is cut [7]. Normally, heterologous expression of proteases in *E. coli* poses issues such as the formation of inclusion bodies [8,9], solubility problems [10], and inactive aggregates [11]. To solve these problems, more complex steps are required, such as adding fusion tags [12,13]. For a thermophilic protease, adaptation of thermophilic proteases over evolution has led to an increased number of disulfide bonds, which are hydrophobic interactions to support their stability at higher temperatures [14]. Obviously, the expression of thermophilic protease in *E. coli* will face more challenges than general protein.

The expression of TTHA0724 in *E. coli* has the problem of low expression and autolysis of the host, which makes it difficult to apply in industry [7]. A novel expression method based on *B. subtilis* RIK1285, which is suited for commercial food-grade production, was developed to solve the issue of poor protein synthesis in *E. coli*. Most signaling peptides use the Sec pathway to direct protease across the cell membrane of *B. subtilis*. TTHA0724 is synthesized in the intracellular precursor form, which has no catalytic activity. The transition from the precursor form of TTHA0724 to the active mature form requires the propeptide to be cut. This is a self-hydrolysis process, with a signal peptide of 20–40 amino acids being broken off before proteases are released outside the cell. Therefore, compared with *E. coli*, *B. subtilis* can conveniently produce more proteases and avoid autolysis problems. Although yeast is one option among numerous heterologous expression hosts, *B. subtilis* provides distinct advantages, such as simplification of downstream purification and strong adaptability for industrial production, as well as meeting food safety standards. In addition, *B. subtilis* has no obvious codon preference, and it does not need to optimize the DNA sequence when expressing heterologous proteins. Tekin et al. successfully generated a serine protease from salt-tolerant *Bacillus* C-125 using *B. subtilis* WB800 [15].

Many strains of *Bacillus* have been utilized to develop proteases for commercial use due to the bacteria’s capacity to produce neutral and alkaline proteases, as well as its robust protein secretion ability [16]. Previous reports have also declared *B. subtilis* and *Bacillus licheniformis* safe for food consumption. Alkaline proteases are widely employed as components in detergents for commercial and domestic use. Although they are in high demand by the detergent industry, alkaline proteases are constrained by their poor activity and instability in the presence of chelating agents, surfactants, oxidizing agents, and bleaching agents [17]. Furthermore, alkaline proteases have become a necessary component rather than a supplementary ingredient, prompting a growing need for proteases that are resistant to surfactants and oxidants. For instance, alkaline protease from *B. clausii* I-52 shows tolerance and stability in presence of anionic surfactant, SDS or oxidants like peroxides [18]. Most of the significant detergent protease additions, such as esperase, savinase, and maxinase, are stable in the presence of various surfactants [19]. Therefore, the search for serine proteases that can remain stable in the presence of surfactants and oxidants is an urgent issue in need of attention.

In this study, a food-grade *B. subtilis* RIK1285 expression system was employed to express the thermophilic protease TTHA0724 from *T. thermophilus* HB8 to avoid the autolysis and inclusion body phenomena observed in *E. coli* [7]. Through screening a signal peptide library, an optimized signal peptide sequence was identified to achieve high-efficiency expression of TTHA0724 and consequently simplify downstream processing. Ultimately, an active form of TTHA0724 was successfully isolated from the culture supernatant and tested for its suitability as a detergent additive and for the generation of soybean polypeptide under high-temperature settings.

## 2. Results

### 2.1. Screening of Signal Peptides

Different signaling peptides can greatly affect the expression level of protein. The aprE signal peptide from the pBE-S is not the best-suited signal peptide for TTHA0724 synthesis. Therefore, 173 signaling peptides from *B. subtilis* Secretory Protein Synthesis System (Takara, Beijing, China) were screened, which covered most types of signaling peptides common to *B. subtilis* protein expression. After electrotransformation of the signal peptide library plasmids into the RIK1285 strain, 2000 colonies were produced. The library was obtained as shown (Figure 1A), with the negative control group containing a few colonies due to the incomplete digestion efficiency of MluI/Eco52I (Figure 1B). Each single clone contains one signal peptide (173 signal peptides, all from the *B. subtilis* family). The control group performed the same operation without signal peptides. Because the efficiency of double digestion did not reach 100%, there were still a few plasmids with aprE that were not completely digested by the enzyme. The results showed that the conversion rate of 173 signal peptides in the experimental group was more than 97%. Sequencing was performed on the five signal peptides with the highest extracellular protein expression (Table 1), and SignalP5.0 was used to predict the cleavage site and secretion mechanism for the same five signal peptides (Table 2).

The activity of TTHA0724 was 2.3 times higher than that of aprE when using yoaW as a signal peptide. Additionally, TTHA0724 produced by RIK1285 did not require intricate purification steps and was only subjected to ultrafiltration and heat treatment, making downstream protease purification more accessible (Figure 2). TTHA0724 resulted in a total of 360 U expressed in *E. coli*. However, a total of 6020 U of TTHA0724 was expressed in *B. subtilis* when using signal peptide yoaW, a 16.72-fold increase (Table 3). Protease activity was determined using Azo casein, which is a casein stained with p-aminobenzenesulfonic acid, and it can be used as a substrate for endoprotease. Azo casein can be used to determine the enzymatic activity of all endoproteases against casein [20].

### 2.2. Characterization of TTHA0724

Bacteria reach the endpoint of growth and development at 24 h, after which they begin to secrete proteases in large quantities. Within 24–48 h, the secretion of protease increased rapidly, but after 48 h, the secretion of protease increase was extremely slow. To achieve the highest efficiency and obtain a large number of proteases, 48 h was selected as the appropriate time for protease synthesis (Figure 3).

Two commercial proteases were used as controls to evaluate the thermostability, temperature, and optimal pH of the thermophilic protease TTHA0724 guided by the yoaW signal peptide (Figure 4). Both TTHA0724 expressed by *B. subtilis* and *E. coli* were shown to retain more than 80% of their activities in a wide temperature range of 65 °C to 95 °C. The optimal temperature of TTHA0724 synthesis in *E. coli* was 75 °C, and the enzyme activity was significantly higher than that at 80 °C. However, when TTHA0724 is expressed by *B. subtilis*, the enzyme activity shows no obvious difference between 75 °C and 80 °C. Therefore, 75 °C is considered the optimal temperature for TTHA0724 (Figure 4a).

With its hyperthermophile characteristics, TTHA0724 can effectively reduce bacterial contamination in various applications, thereby improving manufacturing efficiency. In comparison, the commercial neutral protease and alkaline protease P2000 had activities of over 80% and 90%, respectively, at 40–50 °C and 50–60 °C, respectively. However, a sharp decrease is observed for P2000 after 65 °C, whereas for neutral protease, a steady decrease is observed after 40 °C. After 6 h at 75 °C and 50 °C, TTHA0724 and P2000 still had more than 90% of their original activity, respectively, and the commercial neutral protease retained 80% of its activity after 4 h at 40 °C and 57% after 6 h (Figure 4b). The optimal pH tests revealed that TTHA0724 had the highest level of activity in PB buffer at pH 7.0 while remaining above 80% and 70% at pH 6.0 to 8.0 and 8.0 to 10.0, respectively. The commercial alkaline protease P2000 showed the highest activity in Gly-NaOH buffer at pH 10.0 (Figure 4c).

### 2.3. Application of TTHA0724 in Detergent

Two commonly used proteases—commercial neutral protease and commercial alkaline protease—are known for their high stability; they retain more than 90% of their activity even under conditions of exposure to surfactants and two commonly used bleaching agents. The enzyme activity showed an over 20% increase when expose to 0.5% DMSO, 1% Tween 80, or 1% H_2_O_2_ (Figure 5a). On the contrary, the activity of the original protease in the detergents was inactivated. TTHA0724 retains greater than 90% of its original activity when used in Tide and Ariel liquid detergents (Figure 5b). In fact, it is still unclear exactly why some oxidants increase the activity of subtilin. However, according to some studies, the reason why oxidants enhance protease activity may be through cysteine residues near the active site. This oxidation can lead to conformational changes in the enzyme, thereby increasing its activity. In addition, oxidants can also promote the formation of disulfide bonds between cysteine residues, thus stabilizing the enzyme and increasing its activity [21].

Furthermore, experiments regarding TTHA0724 enzyme addition to dyed fabrics show significant improvement in washing performance, particularly for blood stains, when compared to control experiments with no enzyme (Figure 6). Under the microscope, it was observed that after adding TTHA0724, both blood stains and chocolate stains showed a significant enhancement in the washing effect, and the color attachment on the fabric fibers was reduced (Figure 7). In conclusion, TTHA0724 has great potential as a washing additive; it coexists well with ingredients in detergents such as bleach, and TTHA0724 benefits the removal of protein stains. In particular, the thermal characteristics of TTHA072 show a good killing effect on mites and bacteria on the fabric during high-temperature washing.

### 2.4. Production of Active Peptide from Soybean

In the industry, it is typical to enzymatically hydrolyze soybean meal using neutral and alkaline commercial proteases, with a common amount of 2–5% of the soybean meal. The reaction is carried out at temperatures between 40–55 °C, after which alkaline protease and TTHA0724 can accomplish approximately 23% hydrolysis while neutral protease can only achieve 18%. After 2–3 h, the degree of hydrolysis increased slowly in accordance with the results of previous research (Figure 8a).

A reaction time of 3 h was more economical. At this time, soy protein was quickly digested into peptides, decreasing the amount of substrate. Hydrolysis begins rapidly in 1 h and approaches the highest level in 3 h. Additionally, due to the enzyme’s selectivity, only a small proportion of undivided peptide chains are able to continue the hydrolysis reaction. The products of hydrolysis may compete with protease, decreasing its enzymatic hydrolysis rate. TTHA0724 performed better than commercial alkaline protease P2000, increasing the hydrolysis rate from 20.7% to 23.8% at 75 °C compared to 50 °C (Figure 8b). Higher temperatures break the peptide bonds more effectively and make the soybean meal more soluble in the reaction system. The antioxidant capacity of some peptide liquids generated by hydrolysis of soybean protein with three proteases was measured using DPPH (Figure 8c). TTHA0724 had a DPPH clearance of 53.6% at 75 °C, exceeding that of the other commercial proteases.

TTHA0724 was used to hydrolyze soybean meal, resulting in four different soybean polypeptides of various molecular weights. These polypeptides were found to exhibit significant scavenging ability against peroxy free radicals, as measured by the concentration of one mmol of Trolox equivalent (TE) per gram of dry hydrolysate weight (Figure 9). The ORAC values of the hydrolyzed soybean peptides produced by TTHA0724 ranged from 1.67 to 10.48 mmol TE/g, while those from hydrolysis using neutral protease and P2000 alkaline protease were 0.1 to 0.52 mmol TE/g. According to our research, soybean hydrolysate peptides created with TTHA0724 have displayed a significantly higher ORAC clearance rate than the peptides from the other two commercial proteases. In particular, peptides in the 3–10 kDa range exhibited the most powerful ORAC activity. This finding supports previous studies that have shown soybean peptides to have the most potent physiological activity at approximately 3 kDa [22]. Numerous studies have demonstrated that oligopeptides consisting of 2–20 amino acids generated through soybean hydrolysis possess diverse biological properties, including antioxidant and anti-inflammatory effects. It is worth noting that the majority of soybean peptides have a molecular weight of less than 1000 Da, making up approximately 87.92% of the total. However, peptides with molecular weights ranging from 5–10 kDa make up only a mere 0.05% [23].

While earlier studies suggested that active soybean peptides <3 kDa possess the most potent antioxidant activity, their proportion is relatively lower than that of 3–10 kDa components. It is now evident that active soybean peptides with a molecular weight of 3–10 kDa exhibit the highest antioxidant activity; this suggests that the soybean hydrolysate of TTHA0724 has natural antioxidant potential. It has been observed that low molecular weight hydrolysates exhibit higher peroxyl radical scavenging activity compared to high molecular weight hydrolysates. Furthermore, ultrafiltration is a reliable method of separating various soybean polypeptides, which allows for the convenient acquisition of hydrolyzed polypeptides with varying molecular weights.

## 3. Discussion

### 3.1. The Effect of Signal Peptide and Host on the Synthesis of TTHA0724

The comparative study of acidic, neutral, and alkaline proteases has revealed that extracting alkaline proteases from *B. subtilis* can result in improved efficiency, easy production, and simple genetic manipulation, which is beneficial for the industrial environment. *B. subtilis* is a potential host for recombinant protease production due to its non-pathogenicity and ability to produce extracellular proteases. Therefore, acquiring *B. subtilis* strains capable of producing large amounts of extracellular proteases is of great importance to the industry. For example, employing *B. subtilis* RIK1285 rather than *E. coli* transetta (DE3) can reduce the laborious computations involved in the extraction of protein from bacteria, improve the expression level of proteases, and eliminate the inclusion body phenomenon of proteases in *E. coli*, thus decreasing production costs and simplifying operational steps.

When using *B. subtilis* as the expression host, signal peptides are necessary to guide the protease to the outside of the cell. Therefore, the structure of the signal peptide is closely related to the secretion and synthesis of proteases, and screening of signal peptides provides a basis for realizing the secretion and synthesis of specific enzymes.

The experimental results showed that the expression host had an obvious influence on the synthesis of protein. After replacing *E. coli* with *B. subtilis* RIK1285, the expression levels of TTHA0724 increased by 7.27 times. Selecting a suitable host can solve the problems to some degree, such as low synthesis yield and misfolding of recombinant protein. Moreover, the signal peptide has a significant effect on the secretion and synthesis of proteins. When signal peptide aprE was replaced with yoaW, the expression level of TTHA0724 was increased by 2.3 times, indicating that an appropriate signal peptide can further promote protein secretion.

### 3.2. Enzymatic Characteristics of TTHA0724

For industrial applications, proteases must have high activity and stability under relatively harsh conditions, such as elevated temperatures, a wide range of pH values, and the presence of oxidizing agents and surfactants that affect enzyme activity. Proteases are highly utilized in food, leather, detergents, pharmaceuticals, waste management, and silver recovery, and all industrial proteases must exhibit process suitability and long-term stability. It is especially crucial in the field of leather depilation that proteases exhibit good thermostability, as an increase in temperature can accelerate the hydrolysis reaction rate and thereby save energy in the production process and reduce costs [24].

The expression level of TTHA0724 in *B. subtilis* was 16.7 times higher than that in *E. coli*, and the properties of TTHA0724 were not changed compared with *E. coli*. [7]. Usually, the pH environment of detergents is alkaline, and alkaline proteases are generally better than subtilisin in the detergent industry, with an optimal pH range of 8.5–10.0, consistent with the experimental results. TTHA0724 still maintains more than 80% of its activity at an alkaline pH, benefiting the washing process of typical detergents [25].

### 3.3. Effects of Surfactant, Bleach, and Detergent on TTHA0724

Enzymes in water are affected by different factors; non-aqueous media (such as organic solvents, ionic liquids, supercritical fluids, etc.) provide unique advantages for catalysis, such as changing enzyme selectivity, reversing thermodynamic equilibrium, and avoiding water-dependent side reactions. The activity and stability of enzymes in non-aqueous media are affected by the physical and chemical properties of solvents, such as polarity, functional groups, and molecular structure [26]. Generally, non-polar organic solvents, such as alkanes, have little effect on the secondary and tertiary structure of serine proteases but reduce the enzyme’s flexibility and substrate affinity, thereby reducing the enzyme’s activity. Polar organic solvents, such as alcohols, ketones, nitriles, etc., will form hydrogen bonds or van der Waals forces with serine proteases, affecting the hydration layer and active site of the enzyme, thereby changing the structure and activity of the enzyme [20]. Ionic liquids have significant effects on enzyme activity and stability, mainly depending on the hydrophilic, cation, and anion types of ionic liquids. In general, ionic liquids with low hydrophilicity can maintain the essential water molecules around enzymes, reduce the direct interaction between enzymes and ions, and, thus, enhance the stability of enzymes. Ionic liquids can also change the selectivity of enzymes, mainly depending on the effect of ionic liquids on the water activity in the enzyme microenvironment [27,28].

Previous studies have reported that certain organic solvents can also slightly increase protease activity, e.g., acetonitrile and isopropanol for CAS 5. Methanol, ethanol, ether, hexane, and benzene, on the other hand, have no major effect [29]. Experiments on TTHA0724 demonstrated a 20% increase in enzyme activity in 0.5% DMSO, 1% Tween 80, and 1% H_2_O_2_ conditions, with a small decrease at higher concentrations. These properties can facilitate efficient enzyme engagement in complex solvent environments.

Protease is added to liquid detergent or solid particle washing powder as an additive, but its activity in liquid detergent generally decreases to varying extents. For example, after 93 min of preincubation at 94 °C with 95 mg/mL detergent, RP 1 had only 5% viability remaining in Ariel, 60% in Axion, and 40.7% in Dixan [30]. Similarly, the activity of TTHA0724 diminished in the four detergents, yet more than 80% of the activity remained. Especially in Ariel, which has far better compatibility than that between RP1 and Ariel. Further, the enzyme activity was almost no change in activity when TTHA0724 was in the Unilever and Chaooneng detergents.

### 3.4. The Effect of TTHA0724 on Washing

There is an increasing demand for industrial enzymes that can resist harsh conditions, spurring the discovery of proteases equipped to tackle this requirement. Recent studies have uncovered new proteases, of which thermophilic proteases are found to be more appropriate for challenging industrial conditions than their counterparts operating at normal temperatures and pH values. In the last few years, protease has become one of the main components of detergents, and they can significantly enhance the cleaning effect. Protease itself has a strong ability to break down protein stains. For example, 1000 U LBA 46, when blended with cotton fabric and washed at 2 °C for 40 h, can strip away blood stains with no detergent needed [31].

It is essential to be aware that washing fabrics containing polyester or cotton at high temperatures can negatively impact their structure, resulting in a roughened surface texture, decreased strength, and reduced abrasion resistance. However, such washing may improve the moisture absorption and air permeability of the fabric. To effectively eliminate bacteria and fungi, high-temperature washing is recommended, particularly for seasonal clothing and baby clothes [32]. Temperatures for such treatments typically range between 40 and 60 °C. The study of Mehak Baweja et al. revealed that the protease isolated from the *Bacillus pumilus* MP 27 strain maintained 50% activity at 69 °C [33]. However, TTHA0724 can still maintain more than 90% activity at 60 °C. At 50 °C, its activity drops to approximately 60%, showing better heat resistance. TTHA0724 is suitable for industrial conditions because of its resistance to harsh industrial conditions and compatibility with modern washing machine features, such as high-temperature sterilization.

Although cold-water washing is the most common, high-temperature washing has its advantages. The complex composition of current detergents, such as enzymes used for cleaning, bleach, oxidants, wash aids, and foam suppressors, increases the difficulty of removing stubborn stains. In addition, the phenomenon of stain residue (such as yellowing) can seriously affect the appearance of clothing. These problems are mainly caused by the low cleaning temperature and short cleaning cycle of washing in cold water. Many stains are made up of complex ingredients, including carbohydrates, oils, proteins, and more.

Drum washing machines are equipped with electric heating systems to heat water. Impeller washing machines usually do not have a heating system. In fact, most European washing machines are drum-type and carried out at high temperatures. According to reports, the most commonly used washing temperature in Europe is much higher than in other countries, generally between 40–60 °C. It is generally around 30 °C in North America, while the average washing temperature in Germany is 46 °C. In the United States and South Korea, cold or warm washes up to 40 °C are favored. In contrast, cold washing is more common in China [34,35]. To save energy, washing at low temperatures will be popular. Therefore, while maintaining the stability of TTHA0724, it will be very valuable to improve its activity at low temperatures, especially below 30 °C.

Therefore, in a high-temperature environment, the addition of thermophilic protease can work synergistically with other decontamination components of detergent to better remove colored stains (such as blood stains, colored stains, and sweat stains). Because the presence of thermophilic protease accelerates the washing process, it can effectively shorten the washing time to reduce energy waste and can effectively decompose the residual stains that often remain attached to the fabric to achieve a fast, effective, and energy-saving washing effect.

Additionally, through the application of computer-based molecular modeling, systems biology, and gene editing techniques, thermophilic proteases have the potential to increase their hydrolytic activity at high temperatures, thereby resulting in shorter washing times and more cost savings. Moreover, stabilizers, chemical modifications, and enzyme immobilization can further enhance their stability. Forming composite proteases through the combination of thermophilic proteases and other proteases may create a potent washing effect and provide new solutions for washing product extension.

### 3.5. The Antioxidant Activity of Soybean Peptides Produced by Hydrolysis of TTHA0724

Soybean is an important source of biologically active peptides, which have various beneficial effects on the human body. Finding new soybean peptides with active physiological effects and applications in the food industry is a valuable issue. For instance, antiproliferative effects of soybean peptides on cervical (HeLa, SiHa, CasKi) and breast cancer (MCF7 and MDA-MB-231) cell lines have been reported [36]. In addition, some naturally present peptides in seeds, such as the cholesterol-lowering and anti-inflammatory peptide lunasin from lentils, have shown antioxidant and antihypertensive effects [37]. New fermented soybean food (FSF) created with *B. subtilis* GD1, *B. subtilis* the N4, etc., has antioxidant properties, reducing the accumulation of metabolites, increasing the activity of antioxidant enzymes, and lowering the levels of malondialdehyde (MDA) in serum and liver [38]. With TTHA0724, which is capable of producing active soybean peptides at 75 °C, high temperature may solve contamination and overfermentation problems posed by pathogenic micro-organisms, and it has potential for application in the food industry.

## 4. Materials and Methods

### 4.1. Materials

*T. thermophilus* HB8 was purchased from the Japan Collection of Microorganisms (JCM Japan), while the bacterial strains *E. coli* DH5α, *E. coli* transetta (DE3), *B. subtilis* RIK1285, and the plasmid pBE-S were maintained in our laboratory. The TTHA0724 gene (GenBank No. NC_006461:686268-687572) consisted of 1305 nucleotides and encoded a serine protease comprising 434 amino acids (UniProt accession No. Q5SKB9). Pfu DNA polymerase and protein markers were acquired from TransGen Biotech (Beijing, China), while T4 DNA ligase, restriction endonucleases, the In-Fusion^®^ HD Cloning Plus kit, and Bacteria DNA Extraction Kit, Gel Extraction and Purification Kit, and Plasmid Mini Kit were purchased from Takara (Beijing, China) and Bioteke (Wuxi, China), respectively. Moreover, azo casein (AZO), trichloroacetic acid (TCA), soybean flour, 2,20-azino-bis(3-ethylbenzothiazoline-6-sulfonate) (ABTS), and Trolox were commercially obtained from Sigma-Aldrich (St. Louis, MO, USA). SDS, DMSO, Tween 20, Tween 80, Triton X-100, hydrogen peroxide, sodium perborate, NaOH, phenolphthalein, methanol, and potassium persulfate were purchased from Guoyao (Shanghai, China). DPPH (1,1-diphenyl-2-picryl-hydrazyl radical) was purchased from Sigma (Shanghai, China).

### 4.2. Cloning, Expression, and Purification of TTHA0724

A genomic kit extraction method was used to obtain the genomic DNA of *T. thermophilus* HB8, and the TTHA0724 gene sequence was retrieved from NCBI. The primer design software Primer 5.0 (Premier, Vancouver, BC, Canada) was utilized to design the primers as follows:

Sense primer: 5′-AGTAATGAGCTCCCCCAGACCCCA-3′ and the sense anti-sense primer: 5′-GATTTGGGATCCGGGGAAGCAGTA-3′. Underlined sequences refer to SacI and BamHI restriction sites designed into primers. The PCR amplification procedure was initiated at 98 °C for 5 min, followed by 30 cycles of 98 °C for 10 s, annealing at 55 °C for 5 s, extension at 72 °C for 1 min 25 s, and ending with a 10 min incubation at 72 °C. The DNA fragment was purified using a PCR purification kit. The purified fragment was then digested with SacI and BamHI and inserted into the pBE-S expression vector, which was digested with the same restriction enzymes. Transformation of the pBE-S-0724 plasmid into *B. subtilis* RIK1285 was performed for gene cloning and plasmid amplification.

A single colony containing pBE-S-0724 was incubated in LB liquid medium at 37 °C and 180 rpm for 48 h. The supernatant was centrifuged in a 10,000 kDa ultrafiltration tube at 4000 rpm for 30 min and replaced twice with phosphate buffer (pH 7.0). After heat treatment at 75 °C for 10 min and centrifugation at 12,000 rpm for 30 min, pure enzyme was obtained. The purified protein concentration was quantified using the BCA Protein Assay.

### 4.3. Construction and Screening of a Signal Peptide Library and Synthesis Levels of Different Signal Peptides

An SP DNA mixture of 173 signaling peptides from the *B. subtilis* Secretory Protein Synthesis System (Takara, Beijing) was digested with MLuI and EcoRI and inserted into the plasmid pBE-S-0724. The system without the SP DNA mixture was used as the negative control group. In-fusion (Takara) was used to bind the signal peptide to the double-digested plasmid. The binding reaction system was 4 µL of double digestion products, 1.5 µL of SP DNA mixture, 2 µL of 5× In-fusion HD Enzyme Premix, 2.5 µL of high-purity water, and 15 min at 50 °C. The 2 µL connection product was added into 100 µL *E. coli* HST08, placed in an ice bath for 30 min, heat shocked at 42 °C for 90 s, and placed on ice for 2 min. Then, 900 µL SOC medium (37 °C) was added, it was cultured on a shaking table at 180 rpm at 37 °C for 1 h, centrifuged at 4000 rpm for 3 min, and the supernatant was discarded.

The bacteria were then placed on the plate containing 100 µg/mL ampicillin at 37 °C overnight. The plate was rinsed 3–5 times with 1 mL of LB medium, and all colonies were flushed thoroughly. After collecting the bacteria, the plasmid was extracted, and a recombinant plasmid library containing 173 signal peptides was obtained. Two thousand individual colonies should be obtained to ensure that all 173 signal peptides can be included in the signal peptide library.

The different single colonies containing different signal peptides on the medium plate were individually cultivated into 6 mL LB liquid medium containing 50 µg/mL kanamycin at 37 °C and 180 rpm overnight for culture. Two hundred microliters of overnight culture liquid was added to 200 mL of LB liquid medium containing 50 µg/mL kanamycin and cultured at 37 °C at 180 rpm for 48 h.

The total enzyme activity was detected by checking the supernatant of the above-mentioned LB liquid medium that was cultured for 48 h, using the method in Section 4.4. Enzyme Assay.

### 4.4. Enzyme Assay

The method for determining the protease activity of azo casein, as described by Bezerra et al., was improved [20]. A 1.5 mL Eppendorf Tube containing 0.5% (*w*/*v*) azo casein (100 µL; Sigma), prepared in 0.5 M phosphate buffer, pH 7.0, was incubated with purified protease (300 µL) for 30 min at 75 °C. Then, 400 µL of 10% (*w*/*v*) trichloroacetic acid (TCA) was added to stop the reaction. After 15 min, centrifugation was carried out at 12,000 rpm for 5 min. The enzyme was replaced with a buffer in the control group. The supernatant (200 µL) was added to a 96-well microtiter plate, and the absorbance of this mixture was measured in a microtiter plate reader (TECAN) at 335 nm. The parallel experiment was repeated three times. During hydrolysis of the azo casein substrate, an enzyme activity unit is defined as the enzyme needed to alter the absorbance by 0.001 unit/min.
Enzyme activity (U/mL) = A_335_/(0.001 × T × V)
where T is the reaction time, and V is the milliliters of the enzyme added.

### 4.5. Time Curve of Protein Synthesis

The protease activity at 48 h was set to 100%, and the relative activity at other times was calculated. Each experiment was repeated three times.

### 4.6. Effects of Temperature and pH on TTHA0724 Activity and Stability

The effect of temperature on protease activity was assessed at different temperatures ranging from 65 to 95 °C, 30 to 65 °C, and 30 to 70 °C in increments of 10 °C, using azo casein as the substrate. The relative activity of the enzyme was determined, with the highest activity set to 100%. The thermal stability of the enzyme was evaluated by preincubating purified TTHA0724 at various temperatures. Samples were withdrawn at intervals, with the maximum activity set to 100%.

To determine the optimum pH, the reaction was measured by surveying activity at different pH values (4.5–6.0 in 50 mM HAc-NaAc buffer, 6.5–8.0 in 50 mM phosphate buffer, and 8.5–12.0 in 50 mM Gly-NaOH buffer) in increments of 0.5, using azo casein as the substrate. The relative activity of the enzyme was calculated with the maximum activity set at 100%.

To determine the enzyme pH stability, the protease was incubated in the same buffers with pH values from 4.5 to 12.0 at 75 °C for 30 min. The remaining activity was then measured, with the original activity set to 100%.

### 4.7. Effects of Surfactants and Bleach on TTHA0724

To determine the effects of surfactants and bleach on TTHA0724, azo casein was used as a substrate. The protease was mixed with different concentrations of surfactants (SDS, DMSO, Tween 20, Tween 80, and Triton X-100) and bleach (H_2_O_2_ and sodium perborate). The original activity was set to 100%, and the relative activity of the enzyme was calculated. The residual activity of the enzyme was then detected by incubating the protease in different concentrations of bleach and surfactants at 75 °C for 30 min.

### 4.8. Compatibility of TTHA0724 with Detergent

Six liquid detergents (Unilever, Super Energy, Blue Moon, Liby, Tide, and Ariel) were treated at 100 °C for 1 h to denature the initial protease contained in the detergent. This step was used to ensure that the protease was entirely inactivated, and then TTHA0724 was combined with 1% of the heat-treated detergent. The residual activity of the enzyme was then detected by incubating it in different inactivated detergents at 75 °C for 30 min using azo casein as a substrate. The residual activity of the sample was measured, with the original activity set to 100%.

### 4.9. Effect of TTHA0724 on Washing

The white cotton cloth pieces were cut into a 3 cm × 3 cm square, and 100 μL of pig blood was placed onto it. Then, 0.3 g of chocolate powder was dissolved in 1 mL of water, heated to 80 °C for 10 min to melt it, and then 100 μL was added to the center of the cloth. The stained fabric was placed in a 37 °C incubator and allowed to stand for 20 min to create a blood- and chocolate-stained cloth. To detect the effect of the enzyme TTHA0724 on the washing effect, 480 U of the enzyme and 1% inactivated detergent were filled up to 20 mL with water, and the washing machine environment was simulated at 60 °C and 120 rpm for 20 min. Afterward, a camera and an optical microscope were used to take pictures after rinsing off any surface residues with clean water.

### 4.10. Production of Soybean Active Peptides

After the purchased soybean flour was crushed using a wall-breaking machine, it was sieved with a 100-mesh sieve to remove large particles and then whipped for 10 min to achieve thorough grinding. To analyze the degree of hydrolysis by TTHA0724, a 2.5% concentration of the enzyme was set, with 1500 U of the enzyme added to 50 mL of the complete reaction system. The proteases were then reacted at their respective optimum temperatures in a shaker set at 150 rpm for 30 min, and samples were taken between 1 and 4 h with 1 h intervals. Three different ultrafiltration tubes with intercept molecular weights of 30 kDa, 10 kDa, and 3 kDa were used to separate the components from the enzymolysis solution of soybean meal heated at 75 °C with Tth0724 for 3 h. The enzymolysis solution was centrifuged in a 30 kDa ultrafiltration tube at 4000 rpm for 10 min, and the upper layer of the ultrafiltration tube was collected (>30 kDa components). The lower layer collects were then centrifuged at 10,000 rpm with a 10 kDa ultrafiltration tube for 8 min. The upper layer of the ultrafiltration tube was collected (10–30 kDa components). The lower layer collects were centrifuged at 10,000 rpm for 8 min with a 3 kDa small ultrafiltration tube. The upper layer solutions (3–10 kDa components) and the lower layer solutions (<3 kDa components) were then collected. A BCA protein quantitative kit (Tiangen, Beijing) was used to detect the protein concentration of four components with different molecular weight sizes. The four components were diluted to 2 mg/mL, and the DPPH clearance of each component was detected.

The degree of hydrolysis was detected using the formaldehyde titration method, involving 200 mL of formaldehyde solution to which 12 mL of 0.5% phenol-phthalein solution was added and then titrated with 0.1 M NaOH until a light pink tint was achieved; this produced a neutral formaldehyde solution that was diluted with 5 mL of ultrapure water and 2 mL of the supernatant of the reaction solution. Five drops of 0.5% phenolphthalein solution and 2 mL of the neutral formaldehyde solution were added to this solution, which was then titrated with 0.1 M NaOH until a light pink tint resulted.

The following formula was used to calculate the amino nitrogen content:Amino nitrogen content (mg/mL) = ((V_0_ − V_s_) × 1.4008)/2
where V_0_ is the volume of NaOH consumed by the experimental group, and V_s_ is the volume of NaOH consumed by the blank group.

The content of soybean protein in soybean meal is approximately 43%, and the content of nitrogen in protein is approximately 16%, so the total nitrogen content of soybean meal was calculated. The hydrolysis degree of soybean meal was obtained by dividing the amino nitrogen content by the total nitrogen content of soybean meal, and the time (h)-hydrolysis degree (%) curve was drawn.

### 4.11. DPPH Clearance of Soybean Active Peptides

To measure the antioxidant capacity of active soybean peptides produced by hydrolysis of soybean meal by TTHA0724, a 3 h reaction solution of soybean meal hydrolyzed with 200 μL of protease was added to 400 μL of 0.1 mM DPPH ethanol solution. The dark reaction was conducted for 30 min, and the optical density at 517 nm (OD_517_) was measured. The DPPH clearance was then calculated using the following equation:DPPH clearance rate (%) = (b − a)/b × 100%
where a is the absorbance at OD_517_ of the experimental group, and b is the absorbance at OD_517_ of the protease replaced with absolute ethanol.

### 4.12. Antioxidant Capacity of Soybean Active Peptides

Minor adjustments were made to the TEAC approach employed by Miller et al. [39]. The ABTS stock solution was prepared by mixing 2.45 mM potassium persulfate and 7 mM ABTS solution and left to stand at room temperature for 12–16 h. A pH 7.0 phosphate buffer was then used to obtain a 12-fold dilution of the ABTS stock. In a 96-well microplate, a 20 μL sample of hydrolyzed soybean active peptide was combined with a 200 μL solution of ABTS. Soybean functional peptide samples were replaced with concentrations of Trolox solution (0.15, 0.3, 0.6, 0.9, 1.2, and 1.5 μM), which were then measured spectrophotometrically at 734 nm. The concentration of Trolox and the ABTS clearance were measured in the peptide samples. The results were reported in units of μmol TE/μmol samples.
ABTS approval (%) = (A_0_ − A_s_)/A_0_ × 100%
where A_0_ is the absorbance value of the group that contains the peptide sample, and A_s_ is the absorbance value of the blank group.

## 5. Conclusions

In this study, the thermophilic serine protease TTHA0724 was transferred from an *E. coli* expression vector to *B. subtilis* RIK1285, and a more suitable signal peptide was screened. The autolysis problem in the *E. coli* expression system is solved. Meanwhile, *B. subtilis* RIK1285 does not change the original properties of TTHA0724 compared with expression in *E. coli* [7]. The results revealed that the *B. subtilis* expression system enabled higher TTHA0724 synthesis, leading to an impressive 16.7-fold increase in synthesis. Moreover, TTHA0724 remained substantially active in the pH 6–10 range and retained over 80% of the maximum activity after 6 h of incubation at 75 °C. Additionally, TTHA0724 was found to tolerate surfactants and bleach, and its addition improved the washing effects of detergents. Furthermore, the antioxidant activity of soybean peptides produced by TTHA0724 proved to be much higher than that produced by commercial proteases. Therefore, these results suggest that the thermophilic serine protease TTHA0724 has potential applications as a detergent additive or in the food industry.

## Figures and Tables

**Figure 1 ijms-24-15950-f001:**
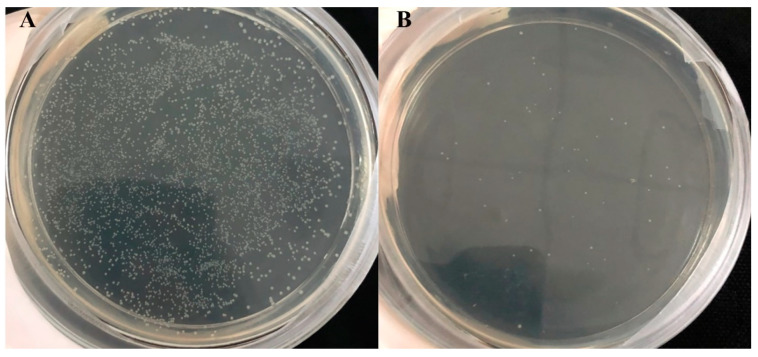
Signal peptide library. (**A**) The SP DNA mixture and (**B**) without the SP DNA combination.

**Figure 2 ijms-24-15950-f002:**
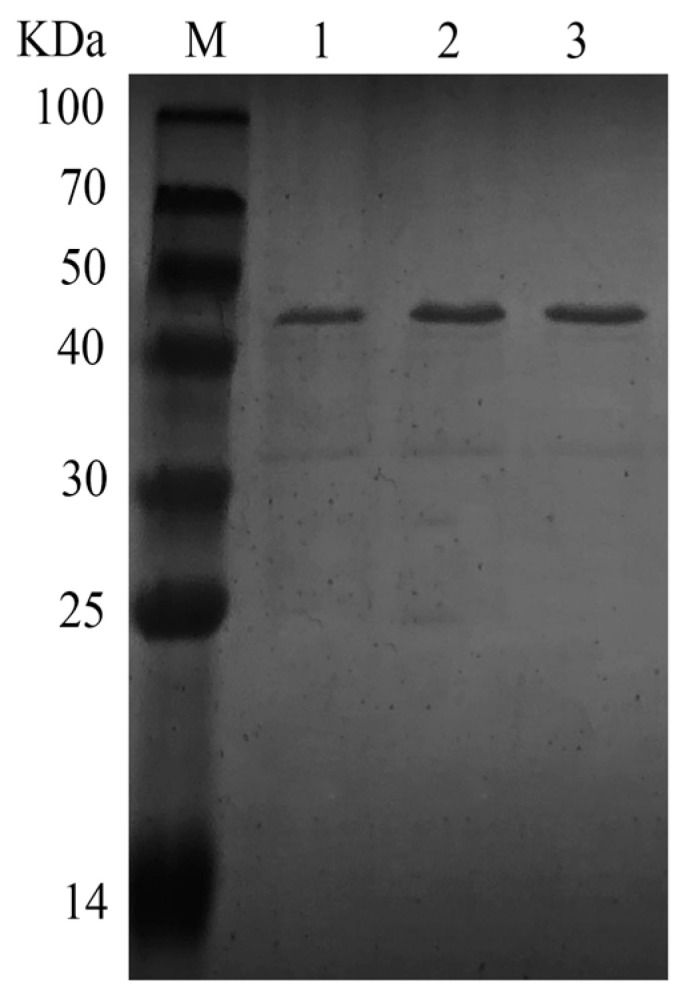
SDS-PAGE of the purified TTHA0724 using different signal peptides expressed in *B. subtilis* revealed a molecular mass of 35 kDa, and its precursor form is 48 kDa (Lane M, molecular weight markers). Lane 1: aprE, Lane 2: yoaW, and Lane 3: ykwD.

**Figure 3 ijms-24-15950-f003:**
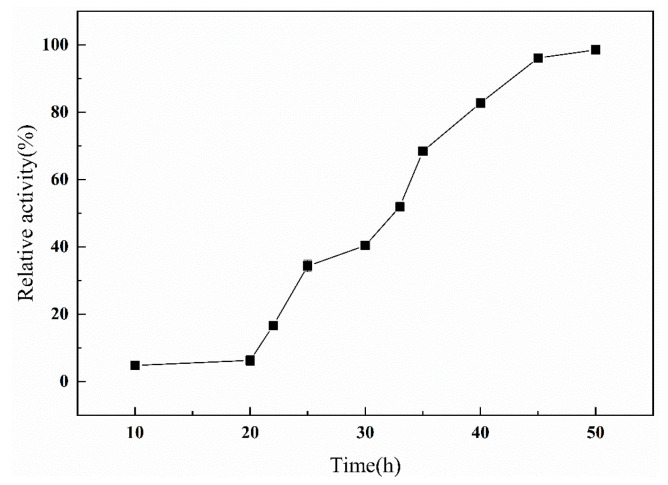
Protein synthesis time curve.

**Figure 4 ijms-24-15950-f004:**
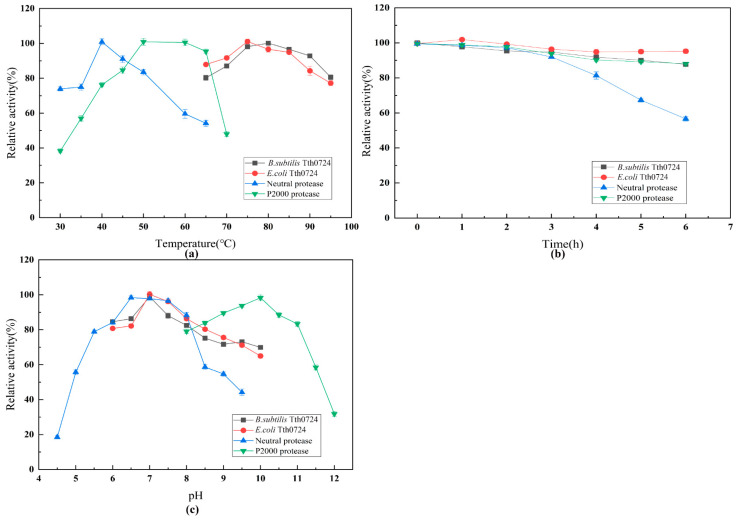
Characterization of different proteases. (**a**) Optimum temperature of proteases; (**b**) thermal stability of proteases. TTHA0724, P2000, and neutral protease were heated at 75 °C, 50 °C, and 40 °C, respectively for 6 h. (**c**) Optimum pH for proteases.

**Figure 5 ijms-24-15950-f005:**
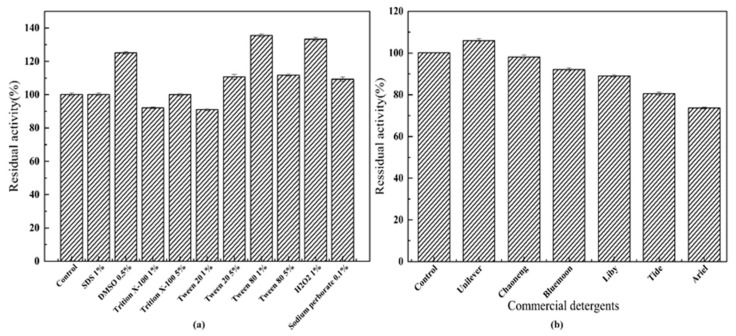
Compatibility of TTHA0724 with detergents and bleaching chemicals: (**a**) compatibility of TTHA0724 with surfactants and bleaches; (**b**) compatibility of TTHA0724 with detergents.

**Figure 6 ijms-24-15950-f006:**
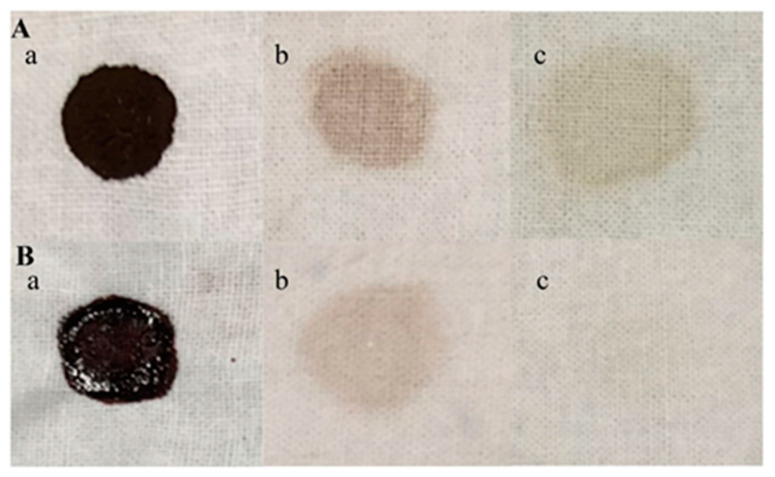
Stained cotton cloth. Group (**A**) is fabric with chocolate stains, including three different treatments (**a**–**c**). Group (**B**) is fabric with blood stains, including three different treatments (**a**–**c**). a is the stain control group, and b is the fabric washed with 1% inactivated detergent. The detergent had been heated to inactivate the protease it contained. c is the fabric washed with 1% protease-inactivated detergent and TTHA0724.

**Figure 7 ijms-24-15950-f007:**
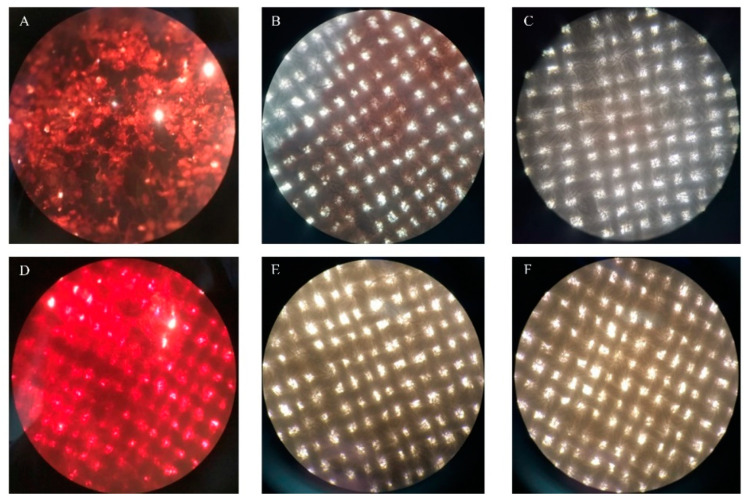
Microscopic washing of stained fabric (40×). (**A**) The fabric with unwashed chocolate stains, (**B**) the fabric after washing chocolate stains with 1% protease-inactivated detergent, (**C**) the fabric after washing chocolate stains with 1% protease-inactivated detergent and TTHA0724; (**D**) unwashed fabric with blood stains, (**E**) fabric after blood stains were washed with 1% protease-inactivated detergent, (**F**) fabric after blood stains were washed with 1% protease-inactivated detergent and TTHA0724.

**Figure 8 ijms-24-15950-f008:**
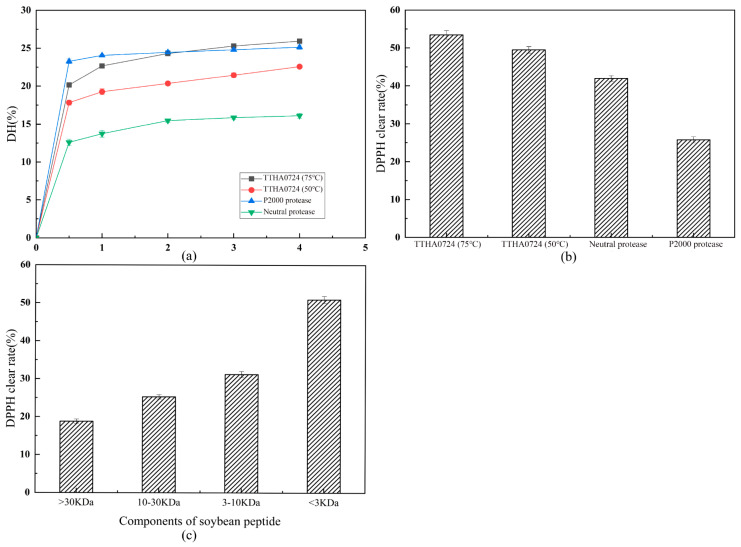
Hydrolysis curve and DPPH clearance of soybean meal by different proteases. (**a**) Hydrolysis curves of TTHA0724 were carried out at 75 °C and 50 °C. Hydrolysis curves of P2000 alkaline protease and neutral protease were carried out at 50 °C and 40 °C, respectively; (**b**) DPPH clearance of soybean peptides produced by hydrolysis of TTHA0724 at 75 °C and 50 °C and by hydrolysis of P2000 protease and neutral protease at 50 °C and 40 °C, respectively; (**c**) DPPH clearance of soybean active peptides with different molecular weights. Soy protein hydrolysates are represented by ORAC activity.

**Figure 9 ijms-24-15950-f009:**
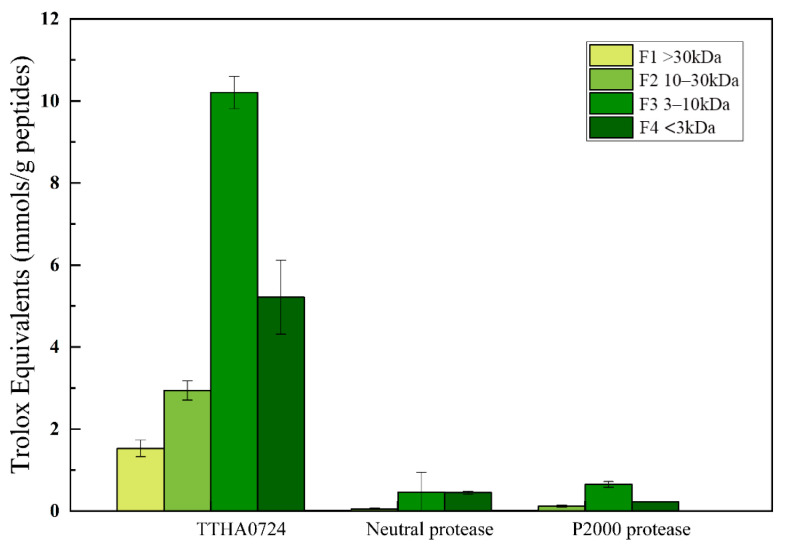
The proteases utilized to hydrolyze soybean protein: TTHA0724, neutral protease, and P2000 alkaline protease. Soy hydrolysates were classified as F1, F2, F3, and F4 (corresponding to samples >30, 30–10, 10–3, or 3 kDa). Each soy hydrolysate fraction was evaluated in triplicate, and ORAC values were expressed as Trolox equivalents per mmol soy hydrolysate (1 mmol TE/G).

**Table 1 ijms-24-15950-t001:** Screened signal peptides.

Signal Peptide	Amino Acid Sequence	Nucleotide Sequence
yurI	MTKKAWFLPLVCVLLISGWLAPAASASA	ATGACAAAAAAAGCATGGTTTCTGCCGCTCGTCTGTGTATTACTGATTTCCGGATGGCTTGCGCCAGCAGCTTCAGCAAGCGCG
yrrS	MSNNQSRYENRDKRRKANLVLNILIAIVSILIVVVA	ATGAGCAATAATCAATCTCGTTATGAAAATCGTGATAAACGCAGAAAAGCCAATTTAGTGCTTAACATTTTAATCGCAATCGTATCCATACT AATTGTCGTAGTAGCAGCG
prE	MRSKKLWISLLFALTLIFTMAFSNMSVQA	ATGAGAAGCAAAAAATTGTGGATCAGCTTGTTGTTTGCGTTAACGTTAATCTTTACGATGGCGTTCAGCAACATGTCTGTGCAGGCT
yoaW	MKKMLMLAFTFLLALTIHVGEASA	ATGAAAAAGATGTTGATGTTAGCTTTTACATTTCTTTTGGCTTTGACTATCCATGTAGGGGAAGCTTCGGCT
ykwD	MKKAFILSAAAAVGLFTFGGVQQASA	ATGAAGAAAGCATTTATTTTATCTGCTGCCGCTGCGGTTGGATTATTCACATTCGGGGGCGTACAGCAAGCATCAGCG

**Table 2 ijms-24-15950-t002:** Signaling peptide secretion pathways and cleavage sites.

Signal Peptide	Secretory Pathway	Cleavage Sites
yoaW	Sec/SPI	Between 22 and 23 residues
ykwD	Sec/SPI	Between 23 and 24 residues
yurI	Sec/SPI	Between 26 and 27 residues
yrrS	Other	Unpredictable
aprE	Sec/SPI	Between 23 and 24 residues

**Table 3 ijms-24-15950-t003:** Expression levels of different host and signal peptides.

Cell	Total Enzyme Activity (U)	Multiple
*E. coli* transetta (DE3)	360 ± 9	1.00
*B. subtilis* RIK1285		
aprE	2620 ± 80	7.27
yoaW	6020 ± 210	16.72
ykwD	5320 ± 390	14.78
yurI	4100 ± 250	11.39
yrrS	5620 ± 410	15.61

## Data Availability

Not applicable.

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
