# Peer review of "Increased Expression Levels of Thermophilic Serine Protease TTHA0724 through Signal Peptide Screening in Bacillus subtilis and Applications of the Enzyme"

_ijms, 2023, doi:10.3390/ijms242115950_

Round 1
Reviewer 1 Report
Comments and Suggestions for Authors
Proteolytic enzymes as detergent additives are essential biocatalysts that help remove a broader range of contaminants in less time, at lower temperatures or in the presence of surfactants. When studying such enzymes, the economic factor is undoubtedly significant. Therefore, it is puzzling why the authors in the paper are looking for a protease that will be active at high temperatures with the possibility of its use in the washing and stain removal processes. This is not clearly explained in this manuscript. This explanation is missing from the description of the planned research. Under these particular conditions mentioned in the text (lines 327-340), isn't high temperature enough without the addition of enzymes? Was there a wash control performed without enzyme addition but at a high temperature?
Please also clarify:
1. what is actually shown in Figure 3, is it the concentration of protein in the culture medium or the proteolytic activity?
2. the caption under Figure 6 is unclear - please clarify what is actually shown in the figures described by letters a, b, and c,
3. the caption under Figure 7 is also unclear; perhaps it would be helpful to present these results in tabular form so that a clear caption follows each figure.
4. is DMSO used as a detergent? (line 294)
Author Response
Dear Professor, thanks for your help to improve our manuscript.
Q1. Proteolytic enzymes as detergent additives are essential biocatalysts that help remove a broader range of contaminants in less time, at lower temperatures or in the presence of surfactants. When studying such enzymes, the economic factor is undoubtedly significant. Therefore, it is puzzling why the authors in the paper are looking for a protease that will be active at high temperatures with the possibility of its use in the washing and stain removal processes. This is not clearly explained in this manuscript.
A: Thanks for the question.
Thermophilic proteases can not only be used for washing, but also for various industrial applications. For example, leather depilation, soybean hydrolysis and casing hydrolysis.
In recent years, it has been increasingly common for washing machines to use high temperatures. 95% of washing machines in North America are vertical axis machines, but the average washing temperature is reported to be about 30 °C (Harrel 2003). According to a study of washing residue and practices in Turkey (Togay 2002), Turkish households wash more than 50% of their clothing at water temperatures above 75°C, with 25% of households washing at temperatures above 85°C. In Germany, for example, the average washing temperature is 46 °C, and only 6% of washes exceed 60°C (Stamminger and Goerdeler 2007).
In general, users who wash at high temperatures tend to use drum washing machines because drum washing machines are usually equipped with an electric heating system to heat water, while impeller washing machines are not (Park et al., 2002). Therefore, it is not surprising that European washing machines are mostly drum type, carried out at high temperatures, and consume the most electricity per wash cycle (and per kilogram capacity); It is reported that the most commonly used washing temperature in Europe is 40 to 60°C (Laitala et al., 2012), which is higher than in other countries.
Cold washing, by contrast, is more common In the United States, China and South Korea, cold or warm washes up to 40°C are favored (Pakula and Staminger, 2010). Interestingly, Chinese drum washing machines seem to use more electricity per cycle (and per kilogram) than South Korea or the United States, regardless of general cold water washing. This may be partly due to China's less stringent energy efficiency standards (compared to South Korea), which allow manufacturers to develop machines with lower energy efficiency.
So, we think the study of high-temperature washing enzymes is a certain need.
In addition, the above reasons are added in 3.4. The effect of TTHA0724 on washing to avoid misunderstanding by readers. For the above reasons, a corresponding quotation is also added to this paragraph. (Line 359-366)
Reference:
- Harrel, C. W. (2003). The US Laundry Market. IEC SC 59D WG Meeting, Gaithersburg, Oct 13, Procter & Gamble communication to IEC SC59D working groups (private communication)
- Togay, M. (2002). Laundry in CEEMEA. Procter & Gamble communication to IEC SC59D working groups (private communication)
- Stamminger, R., & Goerdeler, G. (2007). Aktionstag Nachhaltiges Waschen—Was macht der Verbraucher?
- Pakula, C., Stamminger, R. Electricity and water consumption for laundry washing by washing machine worldwide. Energy Efficiency 3, 365–382 (2010).
- Pakula, C., & Staminger, R. (2010). Electricity and water consumption for laundry washing by washing machine worldwide. Energy Efficiency, 3, 365–382.
- Kim, J., Park, Y., Yun, C. et al. Comparison of environmental and economic impacts caused by the washing machine operation of various regions. Energy Efficiency 8, 905–918 (2015). https://doi.org/10.1007/s12053-015-9333-7
Q2. This explanation is missing from the description of the planned research. Under these particular conditions mentioned in the text (Line 327-340), isn't high temperature enough without the addition of enzymes? Was there a wash control performed without enzyme addition but at a high temperature?
A: Thanks for your suggestion.
In Figure 6, A-b and B-b are the effect drawings of washing with 1% inactivated detergent at 50℃, and it can be clearly seen that there are stain residues. Therefore, just treating blood and chocolate stains with high temperature and detergent will not completely remove them. However, the washing effect was obviously improved after adding protease.
Q3: Please also clarify:
- what is actually shown in Figure 3, is it the concentration of protein in the culture medium or the proteolytic activity?
A: Thank you for the comment.
It's the hydrolytic activity of the protease. The protease activity at 48h was 100%, and the relative activity at other times was calculated. The ordinate is expressed by the relative activity of protease, which can simply indicate the concentration of protease.
The description is added in 4. Materials and Methods (4.5. Protein synthesis time curve).
- the caption under Figure 6 is unclear - please clarify what is actually shown in the figures described by letters a, b, and c,
A: Thanks for your comment.
We have added the revised description below Figure 6.
- the caption under Figure 7 is also unclear; perhaps it would be helpful to present these results in tabular form so that a clear caption follows each figure.
A: Thanks for your comment.
The previous description is unclear for the sake of brevity.
The specific meaning of each image has been added under Figure 7.
- is DMSO used as a detergent? (Line 294)
A: Thanks for the question.
We discuss the compatibility of surfactants, oxidants, and detergents with proteases in part 3.3.
Among them, DMSO is a hydrogen bond-breaking agent, and the compatibility of DMSO and protease is measured because protease is widely used in industry. In order to test whether the protease can adapt to various complex solvent environments.

Reviewer 2 Report
Comments and Suggestions for Authors
The manuscript by Xu et al. is titled “Increased expression levels of thermophilic serine protease Tth0724 through signal peptide screening in Bacillus subtilis and its applications”. As expected from the title, the authors report using a new signal peptide (yoaW) to improve the expression of the Thermus thermophilus serine protease TTHA0724 in Bacillus subtills as the host. However, the focus of the paper is surprisingly broad, including characterization of the enzyme, which makes the title misleading. If such characterization was enabled by the new expression system, since sufficient protein could not previously be obtained, then this approach would make sense. However, the authors were previously able to express the protease in E. coli in sufficient amounts for characterization, including determination of the pH optimum, temperature stability, and relative activities on a number of protein substrates. It is therefore surprising that this kind of characterization is again reported, without proper mention or citation of the 2019 paper (https://doi.org/10.1016/j.ijbiomac.2019.07.101). While this paper is in the reference list, it is not prominently mentioned in the results section, which may be misleading. If the paper is focused on this protease in general, please compare results carefully to previously published literature and change the title to reflect the contents of the paper more carefully.
Please also note that the name of the protein is spelled and typed differently in the title (Tth0724) and in the rest of the manuscript (TTHA0724).
Generally, I would say that the paper is hard to understand without first having read the methods section. Furthermore, the methods section is not perfectly clear and, for example, mentions chemicals and assays that are not mentioned in the main paper (e.g., ABTS and azo casein). I think the authors should rewrite the paper to be interpretable without separately reading the methods section; the results and methods are often more carefully integrated with each other than is the case in this manuscript. For example, it only becomes clear on page 12 that a 173-member signal peptide library from Takara was used, while earlier it was unclear why the authors mixed such a small number of signal peptides to make a library.
Below is a very limited number of detailed comments, I did not comment on all aspects. The manuscript would benefit from careful proofreading and language editing.
Line 24; The introduction contains almost no information on the TTHA0724 from T. thermophilus HB8. Why is this protein the focus of the investigation? What makes it interesting compared to the other proteases mentioned (savinase, esperase, and maxinase)? Is it more resistant to surfactants and oxidants? Since it is from a thermophile one might imagine that the protein is thermostable, but relevant information is not provided in the introduction. What about the activity of the protein at low temperatures (like room temperature), which is very relevant to saving energy as washing machines do not need to be heated? Why is reference 6 only mentioned once, and not in the context of the properties of the protease? It is not clear at all from the introduction why this particular protease is being investigated.
Lines 28-33; The discussion goes from proteases to proteins to transmembrane proteins to recombinant proteins in general. I think a specific focus on factors affecting protease expression problems would be more informative.
Lines 33-35; “However, the recombinase must be cut and heated to…” What does this refer to?
Lines 36-37; “or become highly toxic” Please provide examples and citations; relevance not clear
Line 38; remove “and hydrophobic residues”
Line 42; The Sec pathway is the major pathway
Line 49; What does “and lack of requirement for different codons” mean?
Lines 50-52; Neither the host nor the protein described make sense to mention here
Lines 74-75; This sentence is not necessary here. Since the paper is about the expression system, the reader assumes the protease is already known to be of value, please cite relevant literature (e.g., 10.1016/j.ijbiomac.2019.07.101).
Lines 81-82; What does “Additionally, the pUB ori and Kan resistance genes from pUB110 are present in B. subtilis” mean?
Line 86; Here it would be very valuable to describe how the signal peptide library was constructed; leaving this detail (that a kit from Takara was used) for the end causes confusion
Line 96; Table 1 does not show any data on the levels of extracellular protein expression. How was the library screened? In Section 4.3 it is not clear how many colonies were picked, or why transformants were first scraped from plates to isolate plasmid from this pool (the library). Was this plasmid transformed into a new strain for expression? Page 13 starts with “Single colonies containing different signal peptides on the medium plate were selected” but the authors did not describe the formation of these colonies (e.g., which strain was transformed?).
Table 2; What are bits?
Figure 2; Comparison to the protein expressed in E. coli would be valuable
Table 3; What does “activity of cell extracts in 200mL LB medium” mean? Was the protein not secreted into the medium?
Line 124; Table 2 claims that the cleavage site is known, so it is not correct to state that the purified protease carries the yoaW signal peptide.
Line 130; Figure 4A; From the data presented it cannot be said that there is any significant difference between 75 °C and 80 °C.
Line 154; Please explain that the protease in the detergents was inactivated, otherwise this is confusing (I know it is mentioned in the methods section, but the information is important here).
Line 156; Nowhere in the paper is an explanation given for how hydrogen peroxide treatment can increase the activity of the protease
Figure 6; This was very confusing until I read the methods section. The figures should be independently intelligible, please explain here that the detergent was heat inactivated before use.
Line 179; This title makes it sound like the protease was being hydrolyzed
Line 193; Figure 8B; DPPH clearance rate cannot directly reflect hydrolysis rates (which also cannot be expressed as percentages)
Lines 206-207; No characterization of the size of the hydrolyzed proteins is presented, so how can the authors state that there were four polypeptides of various molecular weights?
Line 215; Please show the data on the peptides produced by the protease. How were peptides separated to measure the antioxidant activities of fractions of different molecular weights?
Lines 252-261; It is not clear what this paragraph is about. It should focus on proteases and protease expression problems/solutions.
Line 270; The enzyme was already characterized in 2019, so a much more detailed comparison to this earlier work is necessary. It is not clear why such basic characterization of the protein is described in a paper dealing with an improved expression system, specifically using an alternative host and signal peptide. The yield was not clearly emphasized to be a problem in the 2019 paper by Xie et al., so more emphasis should be given to that comparison (yield in E. coli vs B. subtilis). The pH preference of the enzyme, for example, has been reported in reference 6, but this is not cited here in the results section.
Lines 288-291; Rigidity cannot explain all the effects of solvents, surfactants, and bleach
Line 295; The increased activity in 1% hydrogen peroxide is very interesting, but discussing the result does not make sense in the context of different solvents. No explanation is offered for why the protein would be more active in hydrogen peroxide.
Line 401; Which plasmid was this?
Line 410; ‘plate containing ampicillin’, not ‘amp resistant plate’
Line 415; This kanamycin concentration is very high, please explain
Line 421; Azo casein is mentioned only in the methods section, please describe its use in the results section as well. This would make understanding the rest of the manuscript simpler.
Lines 426-428; Please explain this control. The protease should be substituted by something inactive like buffer and the rest of the protocol should be the same. The amount of time the azo casein spent in the TCA is much higher in this control. Why is the extra 100 µl 0.5% TCA added to the control?
Line 466; Please state which fabric was used
Line 498; Full name for PDPH must be given
Comments on the Quality of English Language
Manuscript needs to be edited
Author Response
Dear professor, thanks for your great patience and kindness to help us to improve our manuscript.
Q1. The manuscript by Xu et al. is titled “Increased expression levels of thermophilic serine protease Tth0724 through signal peptide screening in Bacillus subtilis and its applications”. As expected from the title, the authors report using a new signal peptide (yoaW) to improve the expression of the Thermus thermophilus serine protease TTHA0724 in Bacillus subtills as the host. However, the focus of the paper is surprisingly broad, including characterization of the enzyme, which makes the title misleading. If such characterization was enabled by the new expression system, since sufficient protein could not previously be obtained, then this approach would make sense. However, the authors were previously able to express the protease in E. coli in sufficient amounts for characterization, including determination of the pH optimum, temperature stability, and relative activities on a number of protein substrates. It is therefore surprising that this kind of characterization is again reported, without proper mention or citation of the 2019 paper (https://doi.org/10.1016/j.ijbiomac.2019.07.101).
A: As you mentioned, the properties of TTHA0724 have been characterized in this article. As the expression host was changed in this article, the recharacterization demonstrated that changing the host merely increased the amount of expression and did not affect the properties of TTHA0724 itself.
Although E. coli can express TTHA0724, However, when the expression level of this protease is high, it leads to cytotoxicity and leads to autolysis. The innovation of this method is that the expression of the active protein is increased, and the autolysis problem is solved because Bacillus subtilis RIK1285 secretes the protein to the outside of the cell.
The use of Bacillus subtilis RIK1285 as a host to express TTHA0724 has the potential for industrial application, The expression of TTHA0724 in Escherichia coli cannot be applied on a large scale.
We cited the 2019 paper; it is the 7th reference in this version of the manuscript (previous version reference 6).
Q2. While this paper is in the reference list, it is not prominently mentioned in the results section, which may be misleading. If the paper is focused on this protease in general, please compare results carefully to previously published literature and change the title to reflect the contents of the paper more carefully.
A: Thanks for your suggestion.
In order to avoid misdirection, we explained this problem and quoted the previous research. (Line 582-584)
The current title might better reflect the main idea of this paper. Although the characterization of TTHA0724 is involved in this paper, it is only to prove that B. subtilis does not change the original properties of TTHA0724, and on this basis, the expression level can be further increased. The characterization of protease properties is not the focus of this article.
The innovation of this paper mainly focuses on the problem of low expression in E. coli and autolysis through signal peptide screening and replacement of the expression system. The combination of the selected signal peptide with Bacillus subtilis can increase the expression level by 16.7 times.
Subsequently, the possibility of TTHA0724 in industrial application was proved by application experiments.
Therefore, we think it is appropriate for the title to focus mainly on the Bacillus subtilis system and the increased expression of signal peptides.
Q3. Please also note that the name of the protein is spelled and typed differently in the title (Tth0724) and in the rest of the manuscript (TTHA0724).
A: Thanks for the comment.
It has been changed in the title.
Q4. Generally, I would say that the paper is hard to understand without first having read the methods section. Furthermore, the methods section is not perfectly clear and, for example, mentions chemicals and assays that are not mentioned in the main paper (e.g., ABTS and azo casein). I think the authors should rewrite the paper to be interpretable without separately reading the methods section;
A: Thanks for your suggestion. In fact, AZO and DPPH are already mentioned in part 4 Materials and Methods.
AZO has been described as a substrate for measuring protease activity. (Line 115-118)
The original paper explicitly mentioned the use of DPPH as a substrate for measuring antioxidants. Because DPPH is a common substrate for measuring antioxidant activity. (Line 213-216, 410-411)
Many parts of this article have been rewritten.
Q5. the results and methods are often more carefully integrated with each other than is the case in this manuscript. For example, it only becomes clear on page 12 that a 173-member signal peptide library from Takara was used, while earlier it was unclear why the authors mixed such a small number of signal peptides to make a library.
A: Thanks for the comment.
In fact, we screened 173 signaling peptides from Bacillus subtilis, which almost covers the common types of signaling peptides expressed by Bacillus subtilis proteins.
This has been explained in detail to avoid misunderstandings. (Line 88-92)
Q6. Below is a very limited number of detailed comments, I did not comment on all aspects. The manuscript would benefit from careful proofreading and language editing.
- Line 24; The introduction contains almost no information on the TTHA0724 from T. thermophilus HB8. Why is this protein the focus of the investigation? What makes it interesting compared to the other proteases mentioned (savinase, esperase, and maxinase)?
A: Thanks for your comment.
Because it has been explained in detail in previous papers published in the laboratory (citation 7), it is not discussed in detail in this paper.
The information about TTHA0724 is mentioned. One of the advantages of TTHA0724 as a serine protease is that it comes from thermophilic bacteria, so it has excellent heat resistance. (Line 34-38)
The necessity of studying serine protease was added in 1. Introduction (Line 25-34).
(2). Is it more resistant to surfactants and oxidants? Since it is from a thermophile one might imagine that the protein is thermostable, but relevant information is not provided in the introduction. What about the activity of the protein at low temperatures (like room temperature), which is very relevant to saving energy as washing machines do not need to be heated?
Thanks for your comment.
Thermophilic enzymes are generally more resistant than normal temperature enzymes and showed good resistance to surfactants and organic solvents.
It is emphasized that TTHA0724 is a thermophilic enzyme. (Line 35-37)
Thermophilic proteases have many important applications, but this article only covers a few of them.
It is increasingly common for washing machines to use high temperatures.
Drum washing machines are usually equipped with an electric heating system to heat water. It is reported that the most commonly used washing temperature in Europe is 40 to 60°C, which is higher than in other countries. Cold washing is more common in China.
Interestingly, Chinese drum washing machines seem to use more electricity per cycle (and per kilogram) than in South Korea or the United States, regardless of general cold water washing. This may be partly due to China's less stringent energy efficiency standards (compared to South Korea), which allow manufacturers to develop machines with lower energy efficiency. This is discussed. (Line 359-366)
In this article, we did not discuss the activity of TTHA0724 at low temperatures, which will be studied in future studies.
(3) Why is reference 6 only mentioned once, and not in the context of the properties of the protease? It is not clear at all from the introduction why this particular protease is being investigated.
A: Thanks for your comment.
Citation 7 has been mentioned in all parts of the article that refer to the previous research. (Line 38, 39, 48, 80, 291, 578)
It has been detailed why is TTHA0724 being studied. (Line 34-37)
(4) Line 28-33; The discussion goes from proteases to proteins to transmembrane proteins to recombinant proteins in general. I think a specific focus on factors affecting protease expression problems would be more informative.
A: Thanks for your suggestion.
It has been described as a common problem encountered in the expression of proteins in E. coli. The B.subtilis expression system can solve these common problems of protease expression in E. coli. (Line 39-42)
Specifically, TTHA0724 in E. coli mainly encountered autolysis and low expression problems. However, the description of transmembrane proteins was deleted.
To sum up, in order to explain the host expression of Bacillus subtilis with signal peptides, which can be a method to solve the problems arising from the expression of proteins in E. coli, an overall discussion of the problems arising from expression in E. coli is necessary.
(5) Line 33-35; “However, the recombinase must be cut and heated to…” What does this refer to?
A: Thanks for the question.
TTHA0724 contains signaling peptides and propeptides, and it can only become an active mature protease after cutting signaling peptides. (Line38-39)
Heat is the wrong expression here. TTHA0724 from T. thermophilus HB8 belongs to the S8 serine protease family and it's a thermophilic enzyme. (Line 34-35)
(6) Line 36-37; “or become highly toxic” Please provide examples and citations; relevance not clear
A: Thanks for your suggestion.
The meaning of this sentence is unclear, and the original meaning is that incorrect folding will cause the protease to lose vitality. The sentence was deleted after considering the context.
(7) Line 38; remove “and hydrophobic residues”
A: Thanks for the suggestion.
These words have been deleted.
(8) Line 42; The Sec pathway is the major pathway
A: Thank you for your suggestion. For accurate description, it has been modified. (Line 51-52)
(9) Line 49; What does “and lack of requirement for different codons” mean?
A: Thanks for the question.
This sentence is not clear, has been modified to "In addition, B. subtilis has no obvious codon preference, and it does not need to optimize the DNA sequence when expressing foreign proteins.” (Line 60-62)
(10) Line 50-52; Neither the host nor the protein described make sense to mention here
A: Thank you for your suggestion.
This sentence has been deleted.
(11) Line 74-75; This sentence is not necessary here. Since the paper is about the expression system, the reader assumes the protease is already known to be of value, please cite relevant literature (e.g., 10.1016/j.ijbiomac.2019.07.101).
A: Thank you for your comment.
This sentence has been deleted and the paper was cited.
(11) Line 81-82; What does “Additionally, the pUB ori and Kan resistance genes from pUB110 are present in B. subtilis” mean?
A: Thank you for your question.
Amp was used to screen the pBE-S shuttle vector in E. coli and Kan was used to screen the pBE-S shuttle vector in Bacillus subtilis.
The cloning method has been explained in detail in the materials and methods, these sentences were deleted.
(12) Line 86; Here it would be very valuable to describe how the signal peptide library was constructed; leaving this detail (that a kit from Takara was used) for the end causes confusion
A: Thank you for your suggestion.
It has been described in detail as the signal peptide library. (Line 88-92)
(13) Line 96; Table 1 does not show any data on the levels of extracellular protein expression. How was the library screened?
A: Thanks for the question.
Table 1 displays the sequence of signal peptides. Table3 shows the protein expression levels of the five most active signaling peptides.
The measurement method of total enzyme activity has been described in detail. (Line 456-458)
(14) In Section 4.3 it is not clear how many colonies were picked, or why transformants were first scraped from plates to isolate plasmid from this pool (the library). Was this plasmid transformed into a new strain for expression? Page 13 starts with “Single colonies containing different signal peptides on the medium plate were selected” but the authors did not describe the formation of these colonies (e.g., which strain was transformed?).
A: Thank you for your suggestion.
2000 single colonies are needed to ensure that the signal peptide library can contain 173 signal peptides. (Line 448-450)
It has stated that all single colonies were cultured in a 6mL LB medium. (Line 465-466)
Detailed instructions have been added. (Line 451-455)
(15) Table 2; What are residue?
A: Thanks for the question.
Bits has been replaced with residue.
(16) Figure 2; Comparison to the protein expressed in E. coli would be valuable
A: Thank you for your suggestion.
Table 3 compares the difference between the expression levels of the two hosts. Figure 2 mainly compares the expression of different signal peptides.
(17) Table 3; What does “activity of cell extracts in 200mL LB medium” mean? Was the protein not secreted into the medium?
A: Thanks for your question.
The sentence was deleted. The measurement method of total enzyme activity is described in detail. (Line 456-458)
(18) Line 124; Table 2 claims that the cleavage site is known, so it is not correct to state that the purified protease carries the yoaW signal peptide.
A: Thanks for the comment.
The text has been changed to the thermophilic protease TTHA0724 guided by yoaW signal peptide. (Line 134)
(19) Line 130; Figure 4A; From the data presented it cannot be said that there is any significant difference between 75 °C and 80 °C.
A: Thank you for your suggestion.
The text has been changed to there is no obvious difference between 75°C and 80°C. (Line 139)
(20) Line 154; Please explain that the protease in the detergents was inactivated, otherwise this is confusing (I know it is mentioned in the methods section, but the information is important here).
A: Thank you for your suggestion.
This part has been edited. (Line 162-164)
(21) Line 156; Nowhere in the paper is an explanation given for how hydrogen peroxide treatment can increase the activity of the protease.
A: Thank you for your suggestion.
This has been added. The literature is also cited to illustrate this problem. (Line 164-170)
(22) Figure 6; This was very confusing until I read the methods section. The figures should be independently intelligible, please explain here that the detergent was heat inactivated before use.
A: Thank you for your suggestion.
It has been modified. (Line 189)
(23) Line 179; This title makes it sound like the protease was being hydrolyzed
A: Thank you for your suggestion.
The title has been changed to “2.4. Produce of soybean active peptide from soybean”.
(24) Line 193; Figure 8B; DPPH clearance rate cannot directly reflect hydrolysis rates (which also cannot be expressed as percentages)
A: Thank you for your comment.
DPPH was used to measure the antioxidant activity of soybean active peptides, not the hydrolytic activity. According to the literature cited, DPPH is a commonly used substrate for measuring antioxidant activity.
(25) Line 206-207; No characterization of the size of the hydrolyzed proteins is presented, so how can the authors state that there were four polypeptides of various molecular weights?
A: Thanks for your question.
A detailed description has been added. (Line 536-550)
(26) Line 215; Please show the data on the peptides produced by the protease. How were peptides separated to measure the antioxidant activities of fractions of different molecular weights?
A: Thank you for the comment.
Methods for separating polypeptides of different molecular weights are described in detail. (Line 522-534)
(27) Line 252-261; It is not clear what this paragraph is about. It should focus on proteases and protease expression problems/solutions.
A: Thank you for your suggestion.
This section has been deleted.
(28) Line 270; The enzyme was already characterized in 2019, so a much more detailed comparison to this earlier work is necessary. It is not clear why such basic characterization of the protein is described in a paper dealing with an improved expression system, specifically using an alternative host and signal peptide. The yield was not clearly emphasized to be a problem in the 2019 paper by Xie et al., so more emphasis should be given to that comparison (yield in E. coli vs B. subtilis). The pH preference of the enzyme, for example, has been reported in reference 6, but this is not cited here in the results section.
A: Thank you for your suggestion.
One purpose of this paper is to demonstrate that changing the expression host does not change the nature of the protease.
The 2019 paper is now cited and further stated that the properties of the protease itself will not be changed on the basis of increasing the expression level. It has been modified. (Line 290-291)
(29) Line 288-291; Rigidity cannot explain all the effects of solvents, surfactants, and bleach
A: Thank you for your comment.
Detailed discussion and references have been added. (Line 297-314)
(30) Line 295; The increased activity in 1% hydrogen peroxide is very interesting, but discussing the result does not make sense in the context of different solvents. No explanation is offered for why the protein would be more active in hydrogen peroxide.
A: Thank you for your comment.
Multiple solvents were discussed together to verify the resistance of TTHA0724 to multiple solvents.
How hydrogen peroxide enhances the activity of protease has been explained. (Line 164-170)
(31) Line 401; Which plasmid was this?
A: Thanks for the question.
The plasmid, which represents the pBE-S-0724 plasmid mentioned in Section 4.1, has been modified. (Line 435-436)
(32) Line 410; ‘plate containing ampicillin’, not ‘amp resistant plate’
A: Thanks for your help.
This issue has been revised. (Line 445)
(33) Line 415; This kanamycin concentration is very high, please explain
A: Thank you for your comment.
- coli and Bacillus subtilis are generally cultured in a medium with a concentration of one part per thousand of antibiotics used in most papers. So, this is a common antibiotic level. The exact number was used to avoid misunderstanding. (Line 451-455)
(34) Line 421; Azo casein is mentioned only in the methods section, please describe its use in the results section as well. This would make understanding the rest of the manuscript simpler.
A: Thank you for your suggestion.
The role of AZO has been explained. (Line 115-118)
(35) Line 426-428; Please explain this control. The protease should be substituted by something inactive like buffer and the rest of the protocol should be the same. The amount of time the azo casein spent in the TCA is much higher in this control. Why is the extra 100 µl 0.5% TCA added to the control?
A: Thank you for your suggestion.
This should be a wrong expression. The control group, in which the enzyme was replaced with a buffer, has been revised in the paper. (Line 465)
(36) Line 466; Please state which fabric was used
Thank you for your comment.
White cotton cloth pieces were used, and it has been revised in paper. (Line 506)
(37) Line 498; Full name for PDPH must be given
A: Thanks for your comment.
DPPH (1,1-diphenyl-2-picryl-hydrazyl radical) was purchased from Sigma (Shanghai, China). The text has been added. (Line 410-411)

Round 2
Reviewer 2 Report
Comments and Suggestions for Authors
Thank you for the detailed response to my comments. I made a few additional remarks, please see the attached pdf.

Author Response
We are very grateful for your help to improve our paper. We have carefully read all the annotations and made modifications according to the suggestions. We believe that your idea about the activity of enzymes at low temperatures is very important. Improving the specific activity of TTHA0724 and its activity at low temperatures is the thing we are now trying to do with site directed mutagenesis. But currently it has only increased by 2.5 times and is not worth publishing yet. Wishing you good health!